# MISLEADER: DEFENDING AGAINST MODEL EXTRACTION WITH ENSEMBLES OF DISTILLED MODELS

## ABSTRACT

Model extraction attacks aim to replicate the functionality of a black-box model through query access, threatening the intellectual property (IP) of machine-learning-as-a-service (MLaaS) providers. Defending against such attacks is challenging, as it must balance efficiency, robustness, and utility preservation in the real-world scenario. Despite the recent advances, most existing defenses presume that attacker queries have out-of-distribution (OOD) samples, enabling them to detect and disrupt suspicious inputs. However, this assumption is increasingly unreliable, as modern models are trained on diverse datasets and attackers often operate under limited query budgets. As a result, the effectiveness of these defenses is significantly compromised in realistic deployment scenarios. To address this gap, we propose MISLEADER (enseMbles of dIStiLled modEls Against moDel ExtRaction) , a novel defense strategy that does not rely on OOD assumptions. MISLEADER formulates model protection as a bilevel optimization problem that simultaneously preserves predictive fidelity on benign inputs and reduces extractability by potential clone models. Our framework combines data augmentation to simulate attacker queries with an ensemble of heterogeneous distilled models to improve robustness and diversity. We further provide a tractable approximation algorithm and derive theoretical error bounds to characterize defense effectiveness. Extensive experiments across various settings validate the utility-preserving and extraction-resistant properties of our proposed defense strategy. Our code is at `anonymous.4open.science/r/misleader-B54B`.

## 1 INTRODUCTION

Machine-learning-as-a-service (MLaaS) platforms have made powerful models widely accessible through simple APIs (Kim et al., 2018; Zhang et al., 2020; Ribeiro et al., 2015), enabling applications in healthcare, finance, and content moderation (ElDahshan et al., 2024; Grigoriadis et al., 2023; Habibi et al., 2024). While this paradigm accelerates AI deployment, it also raises security concerns—most notably, model extraction attacks, where adversaries query a deployed model to train a local replica (Tramèr et al., 2016; Liang et al., 2024b; Kesarwani et al., 2018). Such attacks compromise intellectual property (IP), undermine service value, and pose risks of unauthorized misuse (Pang et al., 2025; Jagielski et al., 2020; Gong et al., 2021; Jiang et al., 2023). As MLaaS adoption continues to expand, developing robust protection mechanisms against model extraction has become a critical requirement for ensuring the security, reliability, and integrity of modern AI systems (Miura et al., 2024; Yan et al., 2022; Li, 2025; Zhao et al., 2025; Cheng et al., 2025).

To address the growing threat of model extraction, recent years have witnessed significant progress in developing defense strategies (Orekondy et al., 2019b; Mazeika et al., 2022; Wang et al., 2023; 2024; Luan et al., 2025). However, a major limitation shared by these existing methods is their reliance on the assumption that attack queries originate from out-of-distribution (OOD) data (Juuti et al., 2019; Liang et al., 2024a). This assumption is motivated by two practical considerations. First, *limited information exposure*: MLaaS APIs typically return only input-output pairs, obscuring the in-distribution (ID) training data available to the attacker (Wang, 2021; Tramèr et al., 2016). Second, *confidentiality measures*: commercial APIs protect their training data to safeguard privacy and intellectual property (Samuelson, 1999; He et al., 2022;?; Borgogno & Colangelo, 2019). These constraints make it appealing to detect or perturb OOD queries based on distributional divergence (Zhang et al., 2023). Yet in practice, such assumptions may fail, as current large-scale models are often trained on vast amounts of diverse, public data (Shen et al., 2024; Lin et al., 2023; Dhamani,

2024), and attackers may possess similarly broad, task-relevant datasets. As a result, distinguishing adversarial queries from benign ones becomes unreliable or infeasible. Moreover, defenses based on the OOD assumption typically adopt a detect–and–divert strategy and, consequently, depend on the presence of OOD inputs. As a result, when these inputs are absent, their protective effect diminishes markedly. For instance, in DNF (Luan et al., 2025), restricting the attacker to in–distribution queries raises the clone accuracy from about $53\%$ to $79\%$, approaching the $\approx 90\%$ attained against an undefended model. Despite this concern, most existing defenses have yet to address model extraction in the absence of OOD assumptions.

In this work, we investigate the novel and critical problem of defending against model extraction without assuming the presence of out-of-distribution (OOD) queries. This is a non-trivial task that introduces several core challenges. In particular, we highlight the following three: (i) *Utility-Robustness Trade-off*. In the absence of OOD indicators, the defender must treat all queries equally—regardless of whether they originate from benign users or adversaries. This requires the defense to preserve accurate predictions for legitimate users while simultaneously degrading the learnability of the model for attackers (Mazeika et al., 2022; Li et al., 2023). Achieving this goal without explicit knowledge of query intent is fundamentally difficult and central to our problem. (ii) *Absence of Attacker Training Information*. In both data-based and data-free model extraction scenarios, attackers may train clone models using surrogate datasets (Tan et al., 2023; Guan et al., 2024) or generate synthetic queries via learned generators (Liu, 2022; Truong et al., 2021; Sanyal et al., 2022). However, the defender has no access to these attacker-specific resources or distributions. This means the defense must be constructed without relying on any information about the attacker's behavior or inputs. (iii) *Theoretical Understanding*. Despite growing empirical interest in model extraction, there is limited theoretical analysis of how defense mechanisms influence extraction risk. A formal foundation is essential to quantify and explain defense effectiveness in real-world scenarios.

To address these challenges, we propose `MISLEADER` (enseMbles of dIstiLled modEls Against moDel ExtRaction), a unified and theoretically grounded defense framework for model extraction. MISLEADER tackles the utility-robustness trade-off by formulating a bilevel optimization objective that preserves fidelity on benign inputs while minimizing the success of potential model clones. To overcome the lack of access to attacker training data, it introduces a data augmentation strategy that approximates attacker queries using transformed variants of the training set, thereby reducing both data-based and data-free threats to a unified and tractable bilevel form. To further strengthen robustness, MISLEADER replaces the single defense model with an ensemble of heterogeneous distilled models, leveraging architectural diversity to increase output variance and disrupt attacker alignment. Finally, we provide theoretical justification through generalization bounds using Rademacher complexity and characterize the extraction risk gap via Wasserstein distance, alongside empirical results validating the effectiveness of MISLEADER across diverse datasets and attack scenarios.

In summary, our contributions are as follows:

- **New Problem Setting for Model Extraction Defense.** We pioneer the study of model extraction defense without relying on OOD assumptions, and formalize a principled mathematical objective that captures the essential trade-offs in this setting. This shift reflects a more realistic deployment environment for MLaaS platforms.

- **Unified Defense with Augmentation and Ensembling.** We propose `MISLEADER`, a practical defense framework that (i) defines a unified bilevel optimization objective for model extraction attacks; (ii) adopts an optimization strategy based on data augmentation; and (iii) ensembles heterogeneous distilled models to boost robustness and preserve utility.

- **Theoretical Guarantees for Generalization and Risk.** We provide a rigorous theoretical foundation for MISLEADER by using Rademacher complexity to bound generalization error, and applying Wasserstein distance to analyze extraction risk under distributional shift. These analyses offer the first formal justification of defenses without OOD assumptions.

The paper is structured as follows. Section 2 formalizes the threat model and model extraction defense setting. Section 3 introduces our proposed novel framework, MISLEADER, by describing the bilevel and data-free trilevel objectives, the unified optimization (Algorithm 1), and the ensemble defense. Section 4 presents the theoretical analysis, including a uniform generalization bound and a Wasserstein-shift bound. Section 5 reports the experimental setup, results, and ablation studies. Section 6 concludes, and the Appendix provides additional details.

## 2 PRELIMINARIES

### 2.1 NOTATIONS

We denote the distribution of the defender's training data as $\mathcal{P}$, and input samples from this distribution as $x \sim \mathcal{P}$. A predictive model is written as $f(x; \boldsymbol{\theta})$, where $\boldsymbol{\theta}$ parameterizes the model, mapping an input $x \in \mathbb{R}^d$ to an output in $\mathbb{R}^k$. The *target model* $f_t(x; \boldsymbol{\theta}_t)$ refers to the deployed black-box model hosted by the MLaaS provider. The *defense model* $d(x; \boldsymbol{\theta}_d)$ is trained by the defender to approximate the target model on inputs from $\mathcal{P}$, while selectively modifying its predictions to hinder model extraction. The attacker constructs queries $x_i \sim \mathcal{Q}$, where $\mathcal{Q}$ denotes the attacker's accessible input distribution, and collects responses $y_i = f_t(x_i)$. Using the resulting labeled set $\{x_i, y_i\}_{i=1}^N$, the attacker trains a *clone model* $f_s(x; \boldsymbol{\theta}_s) \in \mathcal{F}_s$, where $\mathcal{F}_s$ is the hypothesis space accessible to the attacker. We use a bounded loss function $\mathcal{L}(\cdot, \cdot) \leq K$ to quantify the discrepancy between model outputs, such as cross-entropy (with bounded scores) or Jensen–Shannon (JS) divergence.

### 2.2 MODEL EXTRACTION DEFENSE

We consider the model extraction threat in the context of machine-learning-as-a-service (MLaaS), where a deployed target model $f_t(x; \boldsymbol{\theta}_t)$ exposes only black-box access to users. The attacker can submit input queries and observe the model's responses but does not have access to its architecture, training data, or parameters. Let $\mathcal{Q}$ denote the attacker's query distribution. The attacker constructs a query set $\{x_i\}_{i=1}^N$, where each $x_i \sim \mathcal{Q}$, and collects the corresponding outputs $y_i = f_t(x_i)$. These outputs may be soft labels (probability vectors) or hard labels (class indices), depending on the API.

Using the labeled dataset $\{x_i, y_i\}_{i=1}^N$, the attacker trains a *clone model* $f_s(x; \boldsymbol{\theta}_s) \in \mathcal{F}_s$, drawn from a hypothesis class $\mathcal{F}_s$, to approximate the behavior of $f_t$. The model extraction is considered successful if $f_s$ closely mimics $f_t$ under the attacker's input distribution $\mathcal{Q}$. This is formally captured as follows:

**Definition 2.1** (Threat Model). *Let $f_t$ be a target model and $\mathcal{Q}$ the attacker's query distribution. The goal of attackers is to learn a clone model $f_s \in \mathcal{F}_s$ that approximates the behavior of the target:*

$$\min_{f_s \in \mathcal{F}_s} \mathbb{E}_{x \sim \mathcal{Q}} \left[ \mathcal{L}(f_s(x), f_t(x)) \right], \tag{1}$$

*where $\mathcal{L}$ is a task-specific loss function (e.g., cross-entropy or KL divergence).*

Given the above definition of the threat model, we now formally define the model extraction defense. In particular, we seek to construct a defense that proactively mitigates model extraction without requiring knowledge of the attacker's query distribution. This leads to below:

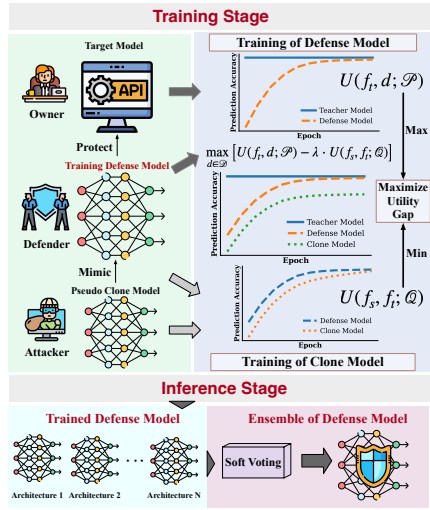

Figure 1: The overview of our proposed MISLEADER framework

**Problem 1** (Model Extraction Defense). *Given a target model $f_t$ trained on a benign data distribution $\mathcal{P}$ and a training dataset $\mathcal{X}_{train} \sim \mathcal{P}$, our goal is to learn a defense model $d \in \mathcal{D}$ that replaces $f_t$ as the deployed black-box model. The defense model must preserve predictive utility on inputs from $\mathcal{P}$, while degrading the effectiveness of any attacker-trained clone model $f_s \in \mathcal{F}_s$ querying $d$, without the assumption that attacker queries originate from out-of-distribution (OOD) sources.*

## 3 METHODOLOGY

In this section, we present our proposed defense framework, MISLEADER, which unifies model extraction defense under a bilevel objective that balances utility for benign users and resistance against extraction across both data-based and data-free settings. We introduce a unified optimization strategy based on data augmentation that approximates attacker queries and enables tractable training without knowledge of the attacker's inputs. Finally, we enhance robustness and utility by ensembling heterogeneous distilled models using soft voting to amplify output diversity and prediction stability. An overview of the framework is shown in Figure 1.

### 3.1 DEFENSE OBJECTIVE FORMULATION

To defend against model extraction, our goal is to construct a defense model $d \in \mathcal{D}$ that preserves utility for benign users while hindering the success of attacker-driven model replication. In this paper, we define utility as the complement of a normalized loss. This design ensures that higher utility corresponds to stronger alignment between models while preserving interpretability in the bounded range $[0, 1]$, facilitating tractable optimization and principled comparisons across models and distributions (Orekondy et al., 2019b; Stanton et al., 2021). The utility function is defined as:

**Definition 3.1** (Normalized Model Agreement Utility). *For any pair of models $f_1$, $f_2$ and input distribution $\mathcal{D}$, we define their agreement utility as*

$$U(f_1, f_2; \mathcal{D}) = 1 - \frac{1}{K}\mathbb{E}_{x \sim \mathcal{D}}\mathcal{L}(f_1(x), f_2(x)),$$

*where $\mathcal{L}(\cdot, \cdot)$ is a bounded loss function with $\mathcal{L}(\cdot, \cdot) \leq K$. This utility quantifies the similarity between model outputs, normalized to lie in $[0, 1]$, with higher values indicating closer alignment.*

Based on the above, we define the *defender's utility* $U(f_t, d; \mathcal{P})$ as the agreement between the defense model $d$ and the target model $f_t$ on the defender's training data $\mathcal{P}$, reflecting prediction consistency for benign users. In contrast, the *attacker's utility* $U(f_s, f_t; \mathcal{Q})$ captures how well the attacker's clone $f_s \in \mathcal{F}_s$ replicates $f_t$ over the attacker's query distribution $\mathcal{Q}$. To balance the competing objectives, the defender solves the following bilevel optimization:

$$\max_{d \in \mathcal{D}} \left[U(f_t, d; \mathcal{P}) - \lambda \cdot U(f_s, f_t; \mathcal{Q})\right], \quad \text{s.t.} \quad f_s = \arg\min_{f \in \mathcal{F}_s} \mathbb{E}_{x \sim \mathcal{Q}}\mathcal{L}(f(x), d(x)), \quad (2)$$

where $\lambda > 0$ controls the trade-off between preserving utility and suppressing extraction.

To make this objective concrete in practical settings, we distinguish between two canonical attacker scenarios that reflect common real-world conditions: *data-based* and *data-free* model extraction. In the former, the attacker queries the model using an external surrogate dataset; in the latter, queries are synthesized using a generator trained from scratch. While both follow the same high-level defense objective, they differ in structure—inducing a bilevel optimization in the data-based case and a trilevel one in the data-free case. We formalize each setting below.

**Data-Based Model Extraction (DBME) Defense.** Here, the attacker has access to a surrogate dataset $\mathcal{X}_{\text{sur}} \sim \mathcal{Q}$, drawn from a distribution potentially different from the defender's data $\mathcal{P}$. The attacker queries the deployed model on $\mathcal{X}_{\text{sur}}$ to collect outputs for training a clone $f_s$. The defender learns $d$ using its own data to preserve utility while suppressing learnability on the surrogate queries:

$$\min_{d \in \mathcal{D}} \max_{f_s \in \mathcal{F}_s} \left[\mathbb{E}_{x \sim \mathcal{P}}\mathcal{L}(f_t(x), d(x)) - \lambda \cdot \mathbb{E}_{x \sim \mathcal{Q}}\mathcal{L}(f_s(x), d(x))\right]. \quad (3)$$

**Data-Free Model Extraction (DFME) Defense.** Here, the attacker synthesizes queries using a generator $g \in \mathcal{G}$ that transforms latent codes $z \sim \mathcal{N}(0, I)$ into query samples $g(z)$. The attacker queries the model on $g(z)$ and uses the responses to train a clone $f_s$. The defender again learns $d$ to maintain fidelity on $\mathcal{P}$ while resisting exploitation via synthetic queries:

$$\min_{d \in \mathcal{D}} \max_{f_s \in \mathcal{F}_s} \left[\mathbb{E}_{x \sim \mathcal{P}}\mathcal{L}(f_t(x), d(x)) - \lambda \cdot \max_{g \in \mathcal{G}} \mathbb{E}_{z \sim \mathcal{N}(0, I)}\mathcal{L}(f_s(g(z)), d(g(z)))\right]. \quad (4)$$

For both scenarios, we design the loss functions to reflect the goals of the defender and the attacker. For the first term, we adopt a knowledge distillation loss that combines the target model output $f_t(x)$ and the ground-truth label $y$, enabling the defense model $d(x)$ to maintain high utility for benign users. For the second term, we use Kullback–Leibler (KL) divergence, as it captures the attacker's objective of imitating the full output distribution of the model, beyond just the predicted class.

### 3.2 UNIFIED OPTIMIZATION STRATEGY

To enable tractable optimization of the defense objective in Section 3.1, we propose a unified strategy based on data augmentation to simulate attacker-facing queries without relying on OOD assumptions. This section is organized into three parts: we first describe the *augmentation pipeline design* that generates diverse proxy queries from the training data, then introduce the *unified optimization objective* that applies to both data-based and data-free settings, and finally detail the *optimization procedure* that jointly updates the defense and attacker models.

---

**Algorithm 1** Unified Optimization of Defense Model Against Model Extraction

---

1: **Input:** Training set $\mathcal{D} = \{(x_i, y_i)\}$; pre-trained target model $f_t$; learning rates $\eta_d$, $\eta_s$; trade-off parameter $\lambda$; number of epochs $N$; batch size $B$; attacker steps $a_{\text{iter}}$; augmentation operator $\mathcal{A}_{\text{aug}}$; $\alpha$; temperature T
2: **Output:** Optimized defense model $d$
3: Initialize defense model $d$ and attacker model $f_s$
4: Generate augmented attacker dataset: $\tilde{\mathcal{X}} = \mathcal{A}_{\text{aug}}(\mathcal{X}_{\text{train}})$
5: **for** epoch $= 1$ to $N$ **do**
6:     **for** each minibatch $\{(x, y)\} \sim \mathcal{D}, \tilde{x} \sim \tilde{\mathcal{X}}$ of size $B$ **do**
7:         **for** $j = 1$ to $a_{\text{iter}}$ **do**                 ▷ Update attacker
8:             $\mathcal{L}^{\text{attacker}} = \mathcal{L}(f_s(\tilde{x}), d(\tilde{x}))$
9:             $f_s \leftarrow f_s - \eta_s \cdot \nabla_{f_s} \mathcal{L}^{\text{attacker}}$
10:         **end for**
11:         Compute defense utility loss: $\mathcal{L}^{\text{defense}} = \mathcal{L}(f_t(x), d(x))$
12:         Recompute attacker loss: $\mathcal{L}^{\text{attacker}} = \mathcal{L}(f_s(\tilde{x}), d(\tilde{x}))$
13:         Total loss: $\mathcal{L}_{\text{total}} = \mathcal{L}^{\text{defense}} - \lambda \cdot \mathcal{L}^{\text{attacker}}$
14:         Update defense: $d \leftarrow d - \eta_d \cdot \nabla_d \mathcal{L}_{\text{total}}$
15:     **end for**
16: **end for**

---

**Augmentation Pipeline Design.** The transformation pipeline includes random resized cropping (for scale and translation), horizontal flipping (for spatial symmetry), affine transformations (for geometric distortion), color jittering (for perturbing brightness, contrast, and saturation), and random grayscale conversion (for reduced color information). These transformations expand the support of the training distribution in directions that are likely to overlap with attacker queries, promoting robustness under both surrogate-driven and generative attacks.

**Unified Optimization Objective.** Let $\tilde{\mathcal{P}} = \mathcal{A}_{\text{aug}}(\mathcal{P})$ denote the augmented input distribution induced by applying the augmentation operator $\mathcal{A}_{\text{aug}}$ to the training distribution $\mathcal{P}$. We formulate a unified defense objective that applies to both data-based and data-free settings:

$$\min_{d \in \mathcal{D}} \max_{f_s \in \mathcal{F}_s} \left[ \mathbb{E}_{x \sim \mathcal{P}} \mathcal{L}(f_t(x), d(x)) - \lambda \cdot \mathbb{E}_{x \sim \tilde{\mathcal{P}}} \mathcal{L}(f_s(x), d(x)) \right], \tag{5}$$

where the first term enforces alignment with the target model on clean data, and the second penalizes extractability based on simulated attacker queries sampled from the augmented distribution.

### 3.3 ENSEMBLES OF DISTILLED MODELS

To improve robustness against model extraction while maintaining utility, MISLEADER replaces the single defense model $d$ with an ensemble of heterogeneous distilled models. This ensemble design is motivated by both empirical and theoretical findings. Prior studies have shown that attackers struggle to replicate target behavior when there is a mismatch in model architectures (Wang et al., 2023; 2024; Luan et al., 2025). Such architectural gaps hinder transferability and reduce alignment with the target's decision boundary (Jagielski et al., 2020; Oliynyk, 2023; Zhao et al., 2023). In parallel, ensemble learning is widely recognized for reducing variance and noise, thus improving prediction accuracy and stability for benign users.

**Training Models with Distillation.** To train each individual defense model in the ensemble, we optimize the unified objective defined in Eq. equation 5. The first term is a distillation loss that encourages the defense model to mimic the target model on benign inputs. Specifically, we follow prior knowledge distillation work (Hinton et al., 2015; Park et al., 2019) and define the loss as:

$$\mathcal{L}^{\text{defense}} = (1 - \alpha) \cdot \mathcal{L}_{\text{CE}}(d(X), Y) + \alpha \cdot T^2 \cdot \text{KL}\left( \text{softmax}\left(\frac{f_t(X)}{T}\right) \| \text{softmax}\left(\frac{d(X)}{T}\right) \right), \tag{6}$$

where $\mathcal{L}_{\text{CE}}$ is the standard cross-entropy loss, and the second term is the KL divergence between softened output distributions. The coefficient $\alpha \in [0, 1]$ balances the contribution of label supervision and teacher guidance, and $T > 0$ is a temperature parameter that softens the probability distributions to provide richer learning signals. The second term in Eq. equation 5, denoted as $\mathcal{L}^{\text{attacker}}$, applies a KL divergence loss to attacker-facing augmented inputs, which penalizes output similarity between

the defense model and the clone model to reduce extractability. We optimize this bilevel objective using an alternating procedure that iteratively updates the attacker and defense models. The training procedure takes a labeled dataset and a pre-trained target model as input, and outputs a single defense model trained to preserve utility while resisting model extraction. The full training algorithm for one model in the ensemble is presented in Algorithm 1.

**Inference with Ensemble of Models.** During inference, MISLEADER aggregates predictions from all ensemble members using soft voting (Kumari et al., 2021). Each model outputs a probability distribution over classes, which are averaged to form the final prediction. This strategy enables independent distillation of diverse models and allows for scalable, parallel deployment without requiring sequential optimization or data partitioning, making it well-suited for our defense setting.

## 4 THEORETICAL ANALYSIS

To support our proposed defense strategy, this section develops theoretical foundations for MIS-LEADER. We aim to address two core questions: $(i)$ how effectively do learned defense and attacker models generalize from finite samples to the true query distribution under data-based extraction, and $(ii)$ how does the distributional shift induced by the generator impact the attacker's ability to replicate the target model under data-free extraction. The resulting bounds spell out when MISLEADER works and where it may fall short.

### 4.1 GENERALIZATION ERROR IN DBME

We first analyze how well the defense model $d \in \mathcal{D}$ and attacker model $f_s \in \mathcal{F}_s$, trained on finite query samples, generalize to the true query distribution $\mathcal{P}$. For analytical tractability, we consider the *aligned* setting $\mathcal{Q} = \mathcal{P}$, which represents the *worst-case scenario* for the defender, as the attacker has full knowledge of the query distribution. This setting enables a clean formulation within the empirical risk minimization framework. When $\mathcal{Q} \neq \mathcal{P}$, the analysis can be extended using Lipschitz arguments Blanchet & Murthy (2019) under additional Lipschitz assumptions, which introduce a Wasserstein-based distributional shift penalty between $\mathcal{P}$ and $\mathcal{Q}$. To control these bounds, we measure model capacity via the *Rademacher complexity*, which quantifies the expressiveness and the generalisation gap of a function class:

**Definition 4.1** (Rademacher Complexity). *Let $\mathcal{G}$ be a class of real-valued functions mapping $\mathcal{X}$ to $[0,1]$. The empirical Rademacher complexity of $\mathcal{G}$ w.r.t. samples $\{x_i\}_{i=1}^n$ is:*

$$\mathfrak{R}_n(\mathcal{G}) := \mathbb{E}_\sigma \left[ \sup_{g \in \mathcal{G}} \frac{1}{n} \sum_{i=1}^n \sigma_i g(x_i) \right],$$

*where $\sigma_i \sim Uniform\{-1, +1\}$ are independent Rademacher variables. Here, the function class is*

$$\mathcal{L}_{\mathcal{D}, \mathcal{F}_s} = \left\{ x \mapsto \mathcal{L}\big(f_t(x), d(x)\big) - \lambda \mathcal{L}\big(f_s(x), d(x)\big) \ \middle| \ \begin{array}{c} d \in \mathcal{D} \\ f_s \in \mathcal{F}_s \end{array} \right\}.$$

**Definition 4.2** (Population and Empirical Risk). *For a defender $d \in \mathcal{D}$ and clone model $f_s \in \mathcal{F}_s$, define the population and empirical risks:*

$$R(f_s, d) := \mathbb{E}_{x \sim \mathcal{P}} \left[ \mathcal{L}(f_t(x), d(x)) - \lambda \mathcal{L}(f_s(x), d(x)) \right],$$

$$\hat{R}(f_s, d) := \frac{1}{n} \sum_{i=1}^n \left[ \mathcal{L}(f_t(x_i), d(x_i)) - \lambda \mathcal{L}(f_s(x_i), d(x_i)) \right],$$

*where $\mathcal{P}$ is the underlying query distribution. Also, define the corresponding minimax risk as:*

$$R_{pop} := \min_{d \in \mathcal{D}} \max_{f_s \in \mathcal{F}_s} R(f_s, d), \quad R_{emp} := \min_{d \in \mathcal{D}} \max_{f_s \in \mathcal{F}_s} \hat{R}(f_s, d).$$

For the generalization analysis, we assume the loss function is bounded, which is a mild assumption that holds for common losses like cross-entropy Mao et al. (2023) or KL divergence when model outputs are bounded. Using this assumption, we establish Theorem 1, which provides a uniform generalization bound for data-based attacks. We further provide detailed proof in the Appendix B.1.

**Assumption 1** (Bounded Loss Function). *$\mathcal{L}$ is upper bounded by a constant $B > 0$.*

**Theorem 1** (Generalization Error for DBME). *Suppose $f_t$ is a fixed target model, and with Assumption 1, then with probability at least $1 - \delta$, the following uniform bound holds:*

$$|R_{pop} - R_{emp}| \leq \sup_{\substack{d \in \mathcal{D} \\ f_s \in \mathcal{F}_s}} \left| R(f_s, d) - \hat{R}(f_s, d) \right| \tag{7}$$

$$\leq 2B\,\mathfrak{R}_n\big(\mathcal{L}_{\mathcal{D}, \mathcal{F}_s}\big) \;+\; B\sqrt{\frac{\log(1/\delta)}{2n}}.$$

Theorem 1 establishes two key generalization results. First, it provides a uniform bound on the generalization error for defender and attacker models, showing that the deviation between the population risk $R(f_s, d)$ and empirical risk $\hat{R}(f_s, d)$ is controlled by the Rademacher complexity of the function class $\mathcal{L}_{\mathcal{D}, \mathcal{F}_s}$ and scales as $\mathcal{O}(1/\sqrt{n})$. Second, it bounds the gap between the population minimax risk $R_{\text{pop}}$ and its empirical counterpart $R_{\text{emp}}$, which implicitly shows how well this empirical solution from finite samples can approximate the optimal solution under true query distributions.

### 4.2 IMPACT OF QUERY DISTRIBUTION SHIFT IN DFME

A key challenge in data-free model extraction is the distributional shift between the attacker's synthetic query distribution $\mathcal{P}_g$, induced by the generator $g$, and the true query distribution $\mathcal{P}$. This shift limits the attacker's ability to replicate the target model $f_t$. To evaluate defense effectiveness, we define *extraction loss* as expected discrepancy between target and clone models over true queries:

$$\mathbb{E}_{x \sim \mathcal{P}} \left[ \mathcal{L}(f_t(x), f_s(x)) \right].$$

A strong defense increases this extraction loss, making it harder for the attacker to succeed. Let $f_s^{\text{DB}}$ and $f_s^{\text{DF}}$ denote the clone models trained under data-based and data-free settings, respectively. Theorem 2 is proposed to quantify the performance degradation caused by this distribution shift and its proof is also deferred to the Appendix B.2.

**Theorem 2** (Generalization Gap for DFME). *Let $\mathcal{L}$ be a loss function that is jointly $\rho$-Lipschitz in both arguments. Suppose the target model $f_t$ and the data-free clone model $f_s^{DF}$ are each L-Lipschitz w.r.t. their input, the generalization gap of the data-free clone model is bounded by:*

$$\left| \mathbb{E}_{x \sim \mathcal{P}} \mathcal{L}\big(f_t(x), f_s^{DF}(x)\big) - \mathbb{E}_{x \sim \mathcal{P}_g} \mathcal{L}\big(f_t(x), f_s^{DF}(x)\big) \right| \tag{8}$$

$$\leq 2\,\rho\,L\,W_1\big(\mathcal{P}, \mathcal{P}_g\big).$$

*where $W_1(\cdot, \cdot)$ denotes the 1-Wasserstein distance.*

This result also provides an intuitive justification for the design of MISLEADER. By amplifying the distribution shift, for example through ensembling heterogeneous models, the generator-induced distribution $\mathcal{P}_g$ may deviate further from the true query distribution $\mathcal{P}$, leading to a larger Wasserstein distance $W_1(\mathcal{P}, \mathcal{P}_g)$. This in turn makes the worst-case scenario more challenging for the attacker, thereby strengthening the defense against model extraction.

## 5 EXPERIMENTS

In this section, we empirically evaluate the effectiveness of the proposed `MISLEADER` framework. Specifically, we aim to address the following research questions: **RQ1**: How effectively does MISLEADER defend against model extraction? **RQ2**: To what extent does MISLEADER preserve serviceability for benign users? **RQ3**: How do model architectures influence defense effectiveness?

### 5.1 EXPERIMENTAL SETUP

**Datasets.** We evaluate the effectiveness of our method against model extraction using MNIST (Deng, 2012), CIFAR-10, CIFAR-100 (Krizhevsky et al., 2009), which are commonly used benchmarks in this line of research. MNIST consists of grayscale images of handwritten digits across 10 classes. CIFAR-10 and CIFAR-100 contain 32×32 natural images with 10 and 100 object categories, respectively.

Table 1: Classification accuracy of clone model on **CIFAR-10** and **CIFAR-100** with *ResNet34* as the target model. Overall, Misleader exhibits the best performance.

| Attack | Defense | CIFAR-10 Clone Model Architecture | | | CIFAR-100 Clone Model Architecture | | |
|---|---|---|---|---|---|---|---|
| | | ResNet18_8X ↓ | MobileNetV2 ↓ | DenseNet121 ↓ | ResNet18_8X ↓ | MobileNetV2 ↓ | DenseNet121 ↓ |
| DFME | Undefended | 87.36 ± 0.78% | 75.23 ± 1.53% | 73.89 ± 1.29% | 58.72 ± 2.82% | 28.36 ± 1.97% | 27.28 ± 2.08% |
| | RandP | 84.28 ± 1.37% | 70.56 ± 2.23% | 70.03 ± 2.38% | 41.69 ± 2.91% | 22.75 ± 2.19% | 23.61 ± 2.70% |
| | P-poison | 78.06 ± 1.73% | 66.32 ± 1.36% | 68.75 ± 1.40% | 38.72 ± 3.06% | 20.87 ± 2.61% | 21.89 ± 2.93% |
| | GRAD | 79.33 ± 1.68% | 65.82 ± 1.67% | 69.06 ± 1.57% | 39.07 ± 2.72% | 20.71 ± 2.80% | 22.08 ± 2.78% |
| | MeCo | 51.68 ± 1.96% | 46.53 ± 2.09% | 61.38 ± 2.41% | 29.57 ± 1.97% | 12.18 ± 1.05% | 10.79 ± 1.36% |
| | ACT | 46.57 ± 2.83% | 40.32 ± 2.96% | 49.25 ± 2.67% | 23.95 ± 2.38% | 10.09 ± 2.53% | 6.26 ± 2.38% |
| | DNF | 53.91 ± 2.30% | 46.32 ± 1.45% | 47.21 ± 2.15% | 18.03 ± 3.03% | 10.82 ± 1.34% | 6.75 ± 1.23% |
| | MISLEADER | **40.70 ± 2.89 %** | **39.82 ± 1.73 %** | **31.73 ± 1.14 %** | **15.28 ± 1.39%** | **7.71 ± 1.65%** | **4.87 ± 1.32%** |
| DFMS-HL | Undefended | 84.67 ± 1.90% | 79.28 ± 1.87% | 68.87 ± 2.08% | 72.57 ± 1.28% | 62.71 ± 1.68% | 63.58 ± 1.79% |
| | RandP | 84.02 ± 2.31% | 78.71 ± 1.93% | 68.16 ± 2.23% | 72.43 ± 1.43% | 62.06 ± 1.82% | 63.16 ± 1.73% |
| | P-poison | 84.06 ± 1.87% | 79.12 ± 1.72% | 68.05 ± 2.17% | 71.83 ± 1.32% | 61.83 ± 1.79% | 62.73 ± 1.91% |
| | GRAD | 84.28 ± 1.96% | 79.36 ± 1.91% | 69.81 ± 1.71% | 71.89 ± 1.37% | 62.60 ± 1.71% | 62.57 ± 1.80% |
| | MeCo | 76.86 ± 2.09% | 71.22 ± 1.87% | 62.33 ± 2.01% | 59.30 ± 1.70% | 55.32 ± 1.65% | 56.80 ± 1.86% |
| | ACT | 73.93 ± 2.67% | 71.97 ± 2.08% | 61.08 ± 2.39% | 55.38 ± 1.97% | 51.46 ± 1.89% | 52.29 ± 2.03% |
| | DNF | 76.51 ± 2.12% | 75.01 ± 1.25% | 61.02 ± 1.21% | 52.98 ± 2.24% | 48.41 ± 1.78% | 49.72 ± 1.24% |
| | MISLEADER | **71.04 ± 2.53%** | **67.46 ± 2.44%** | **60.43 ± 2.10%** | **55.82 ± 1.39%** | **46.03 ± 1.63%** | **45.82 ± 1.51%** |

**Baselines.** We compare SOTA DFME and defense baselines. *Attack Baselines:* (1) Soft-label attack: DFME (Truong et al., 2021). (2) Hard-label attack: DFMS-HL (Sanyal et al., 2022). *Defense Baselines:* We compare to: (1) Undefended: the target model without using any defense strategy; (2) Random Perturb (RandP) (Orekondy et al., 2019b): randomly perturb the output probabilities; (3) P-poison (Orekondy et al., 2019b); (4) GRAD (Mazeika et al., 2022): gradient redirection defense. (5) MeCo (Wang et al., 2023); (6) ACT (Wang et al., 2024); (7) DNF (Luan et al., 2025). Following (Wang et al., 2023), we set the output perturbation budget equal to 1.0 for those defense baselines in the experiments to generate strong defense. That is, $\|\mathbf{y} - \hat{\mathbf{y}}\|_1 \leq 1.0$, where $\mathbf{y}$ and $\hat{\mathbf{y}}$ are the unmodified/modified output probabilities, respectively. This is applied to RandP, P-poison, and GRAD to enhance their defense by inducing more substantial output perturbations. We put more baseline details in Appendix A.4.

**Evaluation Metrics.** We evaluate the performance of MISLEADER across three dimensions to answer the research questions. (1) *Defense Effectiveness:* We measure how well the defense impairs model extraction by evaluating the test accuracy of the clone model trained via black-box access to the defense model. Lower clone accuracy indicates stronger resistance to extraction. (2) *Serviceability:* We assess how well the defense model preserves the predictive utility of the target model by comparing their test accuracies against various defense strategies. Higher test accuracy suggests better predictive utility. (3) *Impact of Model Architectures:* To understand the role of model architecture in defense effectiveness, we compare the clone model accuracy under DFME attacks across various combinations of clone and defense model architectures on CIFAR-10 dataset.

**Implementation Details.** For soft-label attacks, following the setting of (Truong et al., 2021), we set the total number of queries to 2M for MNIST, 20M for CIFAR-10, and 200M for CIFAR-100. For hard-label attacks, we adopt the query budgets used in (Sanyal et al., 2022), setting 8M for CIFAR-10 and 10M for CIFAR-100. Each experiment is repeated five times, and we report the mean and standard deviation of the results. To construct the ensemble model, we first train individual defense models with architectures ResNet18_8x, MobileNetV2, and DenseNet121, and then combine them using soft voting as described in Section 3.3. During training, we employ the attacker in Algorithm 1 with the same architecture as each corresponding defense model. All the experiments are conducted on NVIDIA RTX 6000 GPUs. Additional details are provided in the Appendix A.

## 5.2 DEFENSE PERFORMANCE AGAINST MODEL EXTRACTION

To answer **RQ1** and understand how well Misleader defends against model extraction, we begin by evaluating the clone model's accuracy under both DFME and DBME attacks. This section focuses on quantifying the effectiveness of various defenses in reducing the clone models performance.

**Performance of clone model against DFME.** The results of defense against soft-label and hard-label DFME attack on CIFAR-10 and CIFAR-100 are shown in Table 1. Our backbone is based on ResNet34 (He et al., 2016). We use three distinct model architectures as the clone model architectures, which include ResNet18_8x (He et al., 2016), MobileNetV2 (Sandler et al., 2018), and DenseNet121 (Huang et al., 2017). Compared to the undefended method, under soft-label attack settings, our method can significantly reduce clone model accuracy by 42% to 47% on the CIFAR-

10 dataset and by 21% to 44% on the CIFAR-100 dataset. Under a hard-label attack setting, our method can significantly reduce clone model accuracy by 8% to 14% on CIFAR-10 and around 17% on CIFAR-100. The rest of the results are shown in the Appendix C.2.

**Performance of clone model against DBME.** We evaluate MISLEADER against DBME attacks Knockoff Nets Orekondy et al. (2019a), where the adversary queries the victim model using data drawn from a distribution similar to its training set. Experimental results demonstrate that MIS-LEADER consistently outperforms baseline defenses, achieving improvements exceeding 5% for most cases. Additional details are provided in the Appendix C.1.

Table 2: Evaluation of the utility after applying different defense strategies.

| Method | MNIST ↑ | CIFAR-10 ↑ | CIFAR-100 ↑ |
|---|---|---|---|
| undefended | $98.91 \pm 0.16\%$ | $94.91 \pm 0.37\%$ | $76.71 \pm 1.25\%$ |
| RandP | $98.52 \pm 0.19\%$ | $93.98 \pm 0.28\%$ | $75.23 \pm 1.39\%$ |
| P-poison | $98.87 \pm 0.35\%$ | $94.58 \pm 0.61\%$ | $75.42 \pm 1.21\%$ |
| GRAD | $98.73 \pm 0.31\%$ | $94.65 \pm 0.67\%$ | $75.60 \pm 1.45\%$ |
| MeCo | $98.63 \pm 0.28\%$ | $94.17 \pm 0.56\%$ | $75.36 \pm 0.68\%$ |
| ACT | $98.90 \pm 0.37\%$ | $94.31 \pm 0.75\%$ | $75.78 \pm 0.73\%$ |
| DNF | $98.78 \pm 0.22\%$ | $94.34 \pm 0.07\%$ | $78.73 \pm 0.30\%$ |
| MISLEADER | $\mathbf{98.93 \pm 0.23\%}$ | $\mathbf{95.46 \pm 0.08\%}$ | $\mathbf{78.95 \pm 0.21\%}$ |

## 5.3 Utility of Defense Models

To answer **RQ2**, we measure the test accuracy of the defended model on clean, non-adversarial inputs. Specifically, we assess the utility of the target model under various defense strategies by reporting its test-time performance across different methods in Table 2. As shown, MISLEADER consistently maintains high test accuracy, outperforming state-of-the-art baselines in most cases. Notably, the ensemble of diverse defense models trained under MISLEADER even surpasses the original target model in accuracy across all three architectures. This improvement arises from the ensemble-based knowledge distillation, where multiple distilled models serve as teachers. Such ensembles capture richer predictive behavior and diverse decision boundaries, providing more informative soft targets during training. Combined with ground-truth supervision, this setup promotes smoother generalization and reduces overfitting. These findings aligned with insights from the prior studies Fukuda et al. (2017); Beyer et al. (2022).

## 5.4 Ablation Study: Impact of Model Architectures

To investigate **RQ3**, we first train single defense models with three architectures and attack each of them with three different clone architectures. Figure 2 reports the resulting clone accuracies. Overall, the single models can also beat SOTA baselines in most cases. This verifies the effectiveness of our proposed unified optimization framework. Also, we observe an interesting trend that model extraction succeeds best when the attacker uses the same architecture as the defense, and it weakens when the architectures differ. For instance, a ResNet18_8x defense is copied most accurately by a ResNet18_8x clone, while MobileNetV2 and DenseNet121 clones recover much lower accuracy. This observation aligns with previous insights that the differences of the design principles and architectures would make model stealing harder Liu et al.

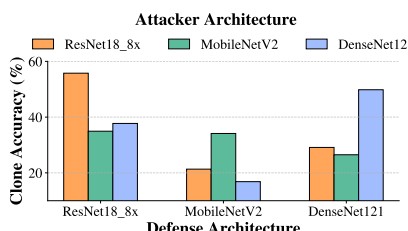

Figure 2: Clone accuracy across different model architectures on CIFAR-10 under DFME attacks. Overall, matching attacker-defender architectures leads to better extraction.

(2022); Wu et al. (2024); Passalis et al. (2020). This further explains the effectiveness of an ensemble, as no single clone model can match every member's architecture, and thus, the effectiveness of defense is much more consistent compared with using only a single defense model.

## 6 Conclusion

In this work, we present MISLEADER, a unified and theoretically grounded defense framework that addresses the model extraction threat in MLaaS settings without relying on OOD assumptions. By formulating a bilevel objective that jointly preserves utility for benign users and degrades extractability, MISLEADER overcomes critical limitations of prior work. Our approach integrates a data augmentation strategy to approximate attacker-facing queries and ensembles heterogeneous distilled models to enhance robustness. Furthermore, we establish theoretical guarantees through generalization bounds and Wasserstein-based risk analysis, offering the first formal justification for model extraction defense under realistic deployment scenarios. Extensive experiments on real-world datasets validate the efficacy of MISLEADER.

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

# A  REPRODUCIBILITY

In this section, we introduce the details of the experiments in this paper for reproducibility. At the same time, we have uploaded all necessary code to our GitHub repository to reproduce the results presented in this paper: `anonymous.4open.science/r/misleader-B54B`. All major experiments are encapsulated as shell scripts, which can be conveniently executed. We introduce the implementation details for reproducibility in the subsections below.

## A.1  REAL-WORLD DATASETS

In this section, we briefly introduce the real-world graph datasets used in this paper, and all these datasets are commonly used datasets in model extraction defense tasks. We present their statistics in Table 3. Specifically, CIFAR-10 includes low-resolution natural images across 10 common object categories such

Table 3: Statistics of the real-world datasets.

| Dataset | # Classes | Input Size | Train/Test Size |
|---|---|---|---|
| CIFAR-10 | 10 | $3 \times 32 \times 32$ | 50k / 10k |
| CIFAR-100 | 100 | $3 \times 32 \times 32$ | 50k / 10k |
| MNIST | 10 | $1 \times 28 \times 28$ | 60k / 10k |

as cats, airplanes, and trucks, and is frequently used to benchmark lightweight models. CIFAR-100 offers a more fine-grained version of CIFAR-10, containing 100 distinct object categories, making it suitable for evaluating the scalability and precision of classification systems. MNIST contains grayscale images of handwritten digits and serves as a canonical benchmark for evaluating robustness and generalization on simple patterns.

## A.2  IMPLEMENTATION OF MISLEADER

This paper implements MISLEADER based on PyTorch (Paszke et al., 2017) and optimizes all models using stochastic gradient descent (Amari, 1993) with momentum and cosine annealing learning rate scheduling (Loshchilov & Hutter, 2016). The training procedure follows a bilevel optimization framework, where the attacker and defender models are alternately updated in each iteration. The defense model is trained using a knowledge distillation loss that combines soft predictions from an ensemble of teacher models and ground-truth labels, using a weighted sum of KL-divergence and cross-entropy. To simulate adversarial queries, we apply aggressive data augmentation to approximate the attacker's query distribution. The distillation temperature $T$, interpolation weight $\alpha$, and attacker regularization coefficient $\lambda$ are treated as tunable hyperparameters. In our experiments, we tune $T$ within the range $[1, 10]$, $\alpha$ within $[0.1, 0.9]$, and $\lambda$ within $[10^{-3}, 10^{-1}]$. To improve training efficiency and stability, we adopt automatic mixed-precision (AMP) (Micikevicius et al., 2017) training with gradient scaling and apply gradient clipping.

## A.3  IMPLEMENTATION OF NEURAL NETWORKS

We implement the ResNet and LeNet model architectures in accordance with prior work (Wang et al., 2023; 2024; Luan et al., 2025), ensuring consistency with established designs for fair comparison and reproducibility. Specifically, ResNet architecture using stacked residual blocks with skip connections, batch normalization, and ReLU activations. The network consists of an initial convolutional layer followed by four sequential residual stages, each downsampling spatial resolution through strided convolutions. The output features are aggregated via global average pooling and passed to a fully connected layer for final prediction. All convolutional layers are initialized using Kaiming normal initialization, and optional input normalization is supported via dataset-specific statistics. The LeNet family is implemented using a standard stack of convolutional layers, ReLU activations, and max-pooling, followed by fully connected layers for classification. The original LeNet5 consists of three convolutional layers with increasing channel sizes (6, 16, 120) and two fully connected layers (84, 10).

## A.4  IMPLEMENTATION OF BASELINES

**RandP** (Orekondy et al., 2019b) adds random noise to the victim model's logits, making it harder to reconstruct true outputs. We adopt its official open-source code[1] for experiments.

---

[1] `https://github.com/tribhuvanesh/prediction-poisoning`

**Prediction Poisoning (P-poison)** (Orekondy et al., 2019b) perturbs predictions by maximizing the angular deviation between original and perturbed gradients to mislead gradient-based clone training. We adopt its official open-source code[2] for experiments.

**GRAD** (Mazeika et al., 2022) redirects gradients to arbitrary directions, disrupting the learning signal for the clone model during extraction. We adopt its official open-source code[3] for experiments.

**MeCo** (Wang et al., 2023) is a model extraction defense method that employs the distributionally robust defensive training. We adopt its official open-source code[4] for experiments.

**ACT** (Wang et al., 2024) uses Bayesian active watermarking to fine-tune the victim model and maximize the model's output change for OOD data. We adopt its official open-source code[5] for experiments.

**DNF** (Luan et al., 2025) employs dynamic early-exit neural networks to introduce increased uncertainty for attackers. We adopt its official open-source code[6] for experiments.

**EDM** (Kariyappa et al., 2021b) ensembles multiple trained networks with a diversity loss to make deliberately dissimilar predictions on OOD inputs. We adopt the original setting in this paper for experiments.

## A.5 IMPLEMENTATION OF THREAT MODELS

**Soft-Label DFME** (Truong et al., 2021): This attack leverages a data generator to synthesize inputs, queries the victim model for soft-labels, and trains the clone model via minimizing KL divergence between victim and clone outputs. This is a soft-label setting, where the black-box API offers softmax probability outputs corresponding to input queries. We adopt the official code [7] for experiments.

**DFMS-HL** (Sanyal et al., 2022): This method uses GANs pre-trained on unrelated classes to generate queries and collects only hard-label predictions from the victim. The generator and clone are trained using classification loss over pseudo-labeled samples. We adopt the official code [8] for experiments.

**Knockoff Nets** (Orekondy et al., 2019a): This method is a data-based model extraction method that extracts the target black-box model using a relevant surrogate dataset to query the target model. Subsequently, the attacker trains a clone model with the surrogate dataset and incorporates the target model predictions on the surrogate dataset as the corresponding data labels. We adopt the official code [9] for experiments.

## A.6 PACKAGES REQUIRED FOR IMPLEMENTATION

We perform all experiments on a server equipped with Nvidia A6000 GPUs. Below we list the key packages and their versions used in our implementation:

- **Python** == 3.11
- **torch** == 2.2.1 + cu121
- **torchvision** == 0.17.1
- **torchaudio** == 2.2.1
- **torchtext** == 0.17.1
- **torch-cuda** == 12.1
- **numpy** == 1.26.4
- **scikit-learn** == 1.1.2

---

[2] https://github.com/tribhuvanesh/prediction-poisoning
[3] https://github.com/mmazeika/model-stealing-defenses
[4] https://github.com/joey-wang123/DFME-DRO
[5] https://github.com/joey-wang123/Bayes-Active-Watermark/tree/main
[6] https://github.com/SYCodeShare/Dynamic-Neural-Fortresses
[7] https://github.com/cake-lab/datafree-model-extraction
[8] https://github.com/val-iisc/Hard-Label-Model-Stealing
[9] https://github.com/tribhuvanesh/knockoffnets

- **scipy** == 1.15.2
- **matplotlib** == 3.8.4
- **pandas** == 2.2.2
- **tqdm** == 4.66.4
- **tensorboard** == 2.16.2
- **protobuf** == 5.29.4

# B  PROOFS

In this section, we provide formal proofs for the theoretical results presented in Section 4.

## B.1  PROOF FOR THEOREM 1

*Proof.* In definition 4.2, we define population and empirical risk as

$$R(f_s, d) := \mathbb{E}_{x \sim \mathcal{P}} \left[ \mathcal{L}(f_t(x), d(x)) - \lambda \mathcal{L}(f_s(x), d(x)) \right], \text{ and}$$

$$\hat{R}(f_s, d) := \frac{1}{n} \sum_{i=1}^{n} \left[ \mathcal{L}(f_t(x_i), d(x_i)) - \lambda \mathcal{L}(f_s(x_i), d(x_i)) \right],$$

where $\{x_i\}_{i=1}^{n} \sim \mathcal{P}$ are i.i.d. samples. We also define their corresponding population and empirical minimax risk as $R_{pop}$ and $R_{emp}$, respectively. Our goal is to bound the quantity

$$|R_{\text{pop}} - R_{\text{emp}}| := \left| \min_{d \in \mathcal{D}} \max_{f_s \in \mathcal{F}_s} R(f_s, d) - \min_{d \in \mathcal{D}} \max_{f_s \in \mathcal{F}_s} \hat{R}(f_s, d) \right|.$$

We can first bound it with:

$$|R_{\text{pop}} - R_{\text{emp}}| \leq \sup_{d \in \mathcal{D}, f_s \in \mathcal{F}_s} \left| R(f_s, d) - \hat{R}(f_s, d) \right|.$$

Since $\mathcal{L}$ is bounded by $B$ by assumption, then by the symmetrization lemma and uniform convergence bound with Rademacher complexity as in Theorem 3.3 of Mohri et al. (2018), we obtain:

$$\sup_{d \in \mathcal{D}, f_s \in \mathcal{F}_s} \left| R(f_s, d) - \hat{R}(f_s, d) \right| \leq 2B \cdot \mathfrak{R}_n(\mathcal{L}_{\mathcal{D}, \mathcal{F}_s}) + B \sqrt{\frac{\log(1/\delta)}{2n}},$$

with probability at least $1 - \delta$, where $\mathfrak{R}_n$ is the empirical Rademacher complexity of the composite function class $\mathcal{L}_{\mathcal{D}, \mathcal{F}_s}$. Hence,

$$|R_{\text{pop}} - R_{\text{emp}}| \leq \sup_{d \in \mathcal{D}, f_s \in \mathcal{F}_s} \left| R(f_s, d) - \hat{R}(f_s, d) \right| \leq 2B \cdot \mathfrak{R}_n(\mathcal{L}_{\mathcal{D}, \mathcal{F}_s}) + B \sqrt{\frac{\log(1/\delta)}{2n}}.$$

$\square$

To tighten the generalization bound, we apply the subadditivity of Rademacher complexity to upper bound the complexity of $\mathcal{L}_{\mathcal{D}, \mathcal{F}_s}$ by the sum over the defense and attack function classes.

**Proposition 1** (Subadditivity of Rademacher Complexity)**.** *Assume the loss function $\mathcal{L}$ is bounded. Then the Rademacher complexity of the composite function class $\mathcal{L}_{\mathcal{D}, \mathcal{F}_s}$ satisfies*

$$\mathfrak{R}_n(\mathcal{L}_{\mathcal{D}, \mathcal{F}_s}) \leq \mathfrak{R}_n(\mathcal{L}_{\mathcal{D}}) + \lambda \, \mathfrak{R}_n(\mathcal{L}_{\mathcal{F}_s}), \text{ where}$$

$$\mathcal{L}_{\mathcal{D}} := \{x \mapsto \mathcal{L}(f_t(x), d(x)) : d \in \mathcal{D}\}, \quad \mathcal{L}_{\mathcal{F}_s} := \{x \mapsto \mathcal{L}(f_s(x), d(x)) : d \in \mathcal{D}, f_s \in \mathcal{F}_s\}.$$

*Proof.* This follows from the sub-additivity of supremum and linearity of expectation.    $\square$

## B.2 PROOF FOR THEOREM 2

*Proof.* Let $f_t$ and $f_s^{\text{DF}}$ be $L$-Lipschitz with respect to their inputs, and suppose the loss function $\mathcal{L}$ is jointly $\rho$-Lipschitz in both arguments. By jointly $\rho$-Lipschitz, we mean that for any inputs $(u_1, v_1)$ and $(u_2, v_2)$, we have $|\mathcal{L}(u_1, v_1) - \mathcal{L}(u_2, v_2)| \leq \rho\left(\|u_1 - u_2\| + \|v_1 - v_2\|\right)$ with $\ell_1$ norm.

Consider two distributions $\mathcal{P}$ (true query distribution) and $\mathcal{P}_g$ (generator-induced distribution). For any $x \sim \mathcal{P}$, let $T(x) \sim \mathcal{P}_g$ be the optimal transport map that minimizes the 1-Wasserstein distance between $\mathcal{P}$ and $\mathcal{P}_g$. Then we can write:

$$\left| \mathbb{E}_{x \sim \mathcal{P}}\left[\mathcal{L}(f_t(x), f_s^{\text{DF}}(x))\right] - \mathbb{E}_{x \sim \mathcal{P}_g}\left[\mathcal{L}(f_t(x), f_s^{\text{DF}}(x))\right]\right|$$
$$= \left| \mathbb{E}_{x \sim \mathcal{P}}\left[\mathcal{L}(f_t(x), f_s^{\text{DF}}(x)) - \mathcal{L}(f_t(T(x)), f_s^{\text{DF}}(T(x)))\right]\right|$$
$$\leq \mathbb{E}_{x \sim \mathcal{P}}\left[\left|\mathcal{L}(f_t(x), f_s^{\text{DF}}(x)) - \mathcal{L}(f_t(T(x)), f_s^{\text{DF}}(T(x)))\right|\right],$$

where the final inequality follows from Jensen's inequality.

By the Lipschitz property of $\mathcal{L}$, and noting that both $f_t$ and $f_s^{\text{DF}}$ are $L$-Lipschitz, we have

$$\left|\mathcal{L}(f_t(x), f_s^{\text{DF}}(x)) - \mathcal{L}(f_t(T(x)), f_s^{\text{DF}}(T(x)))\right|$$
$$\leq \rho \cdot \|f_t(x) - f_t(T(x))\| + \rho \cdot \left\|f_s^{\text{DF}}(x) - f_s^{\text{DF}}(T(x))\right\| \leq 2\rho L \cdot \|x - T(x)\|. \quad (9)$$

Taking expectations:

$$\left| \mathbb{E}_{x \sim \mathcal{P}}\left[\mathcal{L}(f_t(x), f_s^{\text{DF}}(x))\right] - \mathbb{E}_{x \sim \mathcal{P}_g}\left[\mathcal{L}(f_t(x), f_s^{\text{DF}}(x))\right]\right| \leq 2\rho L \cdot \mathbb{E}_{x \sim \mathcal{P}}\left[\|x - T(x)\|\right]$$
$$= 2\rho L \cdot W_1(\mathcal{P}, \mathcal{P}_g).$$

$\square$

## C COMPLEMENTARY EXPERIMENTAL RESULTS

Beyond the main results, we also include complementary experiments—covering DFME on MNIST, DBME on real-world datasets, and query budget analysis—to validate MISLEADER's robustness and utility further. These results highlight its effectiveness across diverse real-world scenarios.

### C.1 DBME DEFENSE

In this subsection, we present additional experimental results regarding the DBME defense on CIFAR-10, CIFAR-100, and MNIST. Table 4 summarizes clone accuracies obtained by Knock-offNets. Overall, MISLEADER delivers the strongest protection on all three datasets, lowering the attacker's accuracy to 40.35%, 54.85%, and 28.76%, respectively. These figures represent at least a five-percentage-point improvement over the best competing baseline in every case (e.g., EDM+ACT on CIFAR-10 and EDM+MeCo on MNIST and CIFAR-100), highlighting MISLEADER's consistent advantage in mitigating DBME.

Table 4: Clone accuracy (%) under KnockoffNets attacks across datasets.

| Baseline | Dataset | | |
|---|---|---|---|
| | MNIST | CIFAR10 | CIFAR100 |
| undefended | 90.18 | 85.39 | 53.04 |
| EDM | 51.34 | 68.50 | 41.16 |
| EDM+MeCo | 46.19 | 59.18 | 35.71 |
| EDM+ACT | 45.78 | 60.21 | 34.67 |
| MISLEADER | **40.35** | **54.85** | **28.76** |

### C.2 DFME DEFENSE ON MNIST

In this subsection, we present additional experimental results regarding the DFME defense on MNIST, as presented in Table 5. The substantial advantage of MISLEADER on MNIST stems

Table 5: Accuracy of the clone model on the MNIST dataset using LeNet5 as the target model.

| Attack | Defense | CIFAR-10 Clone Model Architecture | | |
|---|---|---|---|---|
| | | LeNet5 ↓ | LeNet5-Half ↓ | LeNet5-1/5 ↓ |
| DFME | undefended | $98.76 \pm 0.27\%$ | $96.65 \pm 0.43\%$ | $94.62 \pm 0.69\%$ |
| | RandP | $92.25 \pm 0.32\%$ | $91.86 \pm 0.49\%$ | $90.37 \pm 0.73\%$ |
| | P-poison | $88.34 \pm 0.78\%$ | $86.09 \pm 0.96\%$ | $84.98 \pm 1.07\%$ |
| | GRAD | $87.22 \pm 0.70\%$ | $85.38 \pm 0.91\%$ | $84.23 \pm 1.16\%$ |
| | MeCo | $85.07 \pm 0.87\%$ | $82.93 \pm 1.27\%$ | $82.57 \pm 1.53\%$ |
| | ACT | $81.67 \pm 0.96\%$ | $80.18 \pm 1.38\%$ | $80.09 \pm 1.76\%$ |
| | DNF | $55.97 \pm 3.19\%$ | $42.26 \pm 5.78\%$ | $51.84 \pm 4.27\%$ |
| | MISLEADER | $\mathbf{11.81 \pm 0.64\%}$ | $\mathbf{11.80 \pm 2.98\%}$ | $\mathbf{15.91 \pm 2.97\%}$ |

from its ensemble-based design, which introduces architectural diversity and more complex decision boundaries that resist imitation. Specifically, under a constrained query budget of 2M, MISLEADER reduces clone accuracy to below 16%, which substantially outperforms all other defenses. Notably, even when the query budget is increased tenfold to 20M, the clone model accuracy remains low at $14.09 \pm 1.03\%$ (LeNet5 as clone model), further demonstrating the robustness of our strategy. Overall, all the experiments in this paper highlight that MISLEADER's ensemble distillation amplifies distributional shifts and disrupts attacker generalization.

### C.3 IMPACT OF QUERY BUDGET

To assess the impact of query budget on extraction performance, we experiment with a range of query budgets representing different attacker capabilities. This setup allows us to evaluate the robustness of defense strategies under varying levels of query access. As visualized in Figure 3, MISLEADER consistently outperforms state-of-the-art defenses across all budget levels, maintaining low clone model accuracy even when the attacker is granted a significantly larger number of queries. These results highlight MISLEADER's resilience to stronger adversaries and demonstrate its effectiveness under both low- and high-budget threat scenarios.

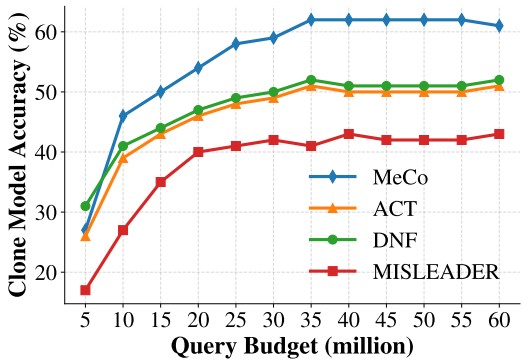

Figure 3: The test accuracy of the clone model (ResNet18_8x) on CIFAR10 with varying query budget. MISLEADER significantly outperforms other SOTA baselines.

## D RELATED WORK

**Model Extraction.** Model extraction (ME) attack aims to extract and clone the functionality of the public API with only query access. Based on the query data used by the attacker, model extraction techniques can be classified into two categories: data-based and data-free model extraction. (1) Data-based Model Extraction (DBME): DBME focuses on extracting the victim model using real dataset (Papernot et al., 2017; Orekondy et al., 2019a; Kariyappa et al., 2021b). (2) Data-free Model Extraction (DFME): DFME, on the other hand, aims to extract the victim model using synthetic data exclusively (Truong et al., 2021; Kariyappa et al., 2021a; Wang, 2021; Sanyal et al., 2022; Hu et al., 2023). These approaches reduce the dataset requirement for stealing the victim model.

**Model Extraction Defense.** Current model extraction defense methods can be broadly categorized into *passive* and *active* strategies. *Active defense* techniques aim to proactively disrupt the extraction process by manipulating model behavior—either through output perturbation, architectural manipulation, or adversarial retraining. These include *output-perturbation-based methods*, such as P-poison (Orekondy et al., 2019b), Adaptive Misinformation (Kariyappa & Qureshi, 2020), and GRAD (Mazeika et al., 2022), which modify predictions to mislead attackers; *ensemble-based defenses* that increase extraction difficulty through architectural heterogeneity (Kariyappa et al.,

2021b); and *defensive training* approaches like MeCo (Wang et al., 2023), which retrain models for improved robustness. Other recent active defenses include ACT (Wang et al., 2024), which introduces Bayesian watermarking to amplify response shifts under OOD queries, and DNF (Luan et al., 2025), which leverages dynamic early-exit networks to inject uncertainty into model responses. In contrast, *passive defense* methods detect or verify model theft post hoc, including *detection-based methods* that identify adversarial query patterns (Juuti et al., 2019), and *verification-based approaches* such as watermarking (Adi et al., 2018) and dataset inference (Maini et al., 2021).

Despite these advancements, a key limitation persists: nearly all existing defenses assume the attacker issues out-of-distribution (OOD) queries, which may not hold in realistic MLaaS deployments where public datasets are diverse and easily accessible. In this work, we challenge this assumption and propose a novel active defense strategy that does not rely on OOD detection. The unique OOD-agnostic design makes MISLEADER uniquely suited for real-world MLaaS scenarios, where defense reliability must persist under unrestricted attacker behavior.

## E    FUTURE WORK

While MISLEADER shows strong performance, it opens several avenues for future improvement. For example, more efforts can be made to extend the framework to more adaptive threat models where the attacker and defender co-evolve their strategies. Also, exploring more efficient approximation methods may reduce training overhead and improve deployment feasibility. Lastly, evaluating MISLEADER on even larger-scale datasets would help further verify its scalability.

