# OpenReview forum: "MISLEADER: Defending against Model Extraction with Ensembles of Distilled Models"
_ICLR.cc/2026/Conference — ICLR 2026 Conference Withdrawn Submission_

### Official Review · Reviewer_mDAU · 2025-10-26

**Soundness:** 3
**Presentation:** 2
**Contribution:** 2
**Rating:** 2
**Confidence:** 4

**Summary:**

The paper introduces a defense against model extraction that employs an ensemble of defense models to replace the target model. These defense models are optimized using a bilevel objective that balances fidelity to the target with resistance to cloning. Theoretical analyses show when the proposed method is effective and where it may fall short. Experiments on MNIST and CIFAR-10/100 demonstrate lower clone accuracies and higher clean accuracy for the defended ensemble compared to several existing defenses.

**Strengths:**

1. Defending against model extraction without relying on OOD-query detection is practical in real-world scenarios.
2. The optimization process is simple and intuitive to implement.

**Weaknesses:**

1. Limited novelty: The core component, as the title suggests, to use an ensemble of models to defend against model extraction is similar to EDM [1]. While the authors claim to use data augmentation to approximate attacker queries, there is no analysis demonstrating that such augmentation truly approximates attacker queries. Moreover, the augmentation techniques used are standard practices in machine learning.
2. Incorrect highlight: In Table 1, under the DFMS-HL attack on CIFAR-100 with the clone model ResNet18_8x, their method is not the best but is incorrectly highlighted as such. The table caption, “Overall, MISLEADER exhibits the best performance”, is therefore misleading.
3. Inconsistent baselines: The authors compare their method with EDM [1] in the appendix, but EDM is not included in the main paper’s comparisons, making the evaluation incomplete.
4. Lack of computational resource analysis: Training multiple attacker and defense models requires significantly more computational resources than other methods. A quantitative comparison of computational cost is necessary to assess practical feasibility, especially for larger images or datasets such as PASCAL.
5. Minor typos: The paper uses the ICLR 2025 LaTeX template instead of the 2026 version. Additionally, in line 051, one reference is incorrectly marked with a “?” placeholder.

[1] Sanjay Kariyappa, Atul Prakash, and Moinuddin K Qureshi. Protecting dnns from theft using an ensemble of diverse models. In International Conference on Learning Representations, 2021b.

**Questions:**

Please refer to my concerns in the weakness.

---

### Official Review · Reviewer_GMs9 · 2025-10-27

**Soundness:** 3
**Presentation:** 3
**Contribution:** 2
**Rating:** 4
**Confidence:** 3

**Summary:**

This paper proposes MISLEADER, a novel framework for defending against model extraction attacks without assuming that attacker queries are out-of-distribution (OOD). The defense is formulated as a bilevel optimization problem that balances utility preservation (for benign users) and robustness against extraction. MISLEADER introduces three core ideas:

1. Data augmentation-based proxy queries to simulate attacker behavior;
2. Ensemble of heterogeneous distilled models to enhance robustness and prediction stability;
3. Theoretical justification through generalization bounds (via Rademacher complexity) and Wasserstein-based analysis for data-free settings.

Extensive experiments on MNIST, CIFAR-10, and CIFAR-100 show that MISLEADER achieves superior defense performance while maintaining high prediction utility compared to state-of-the-art baselines such as GRAD, MeCo, ACT, and DNF.

**Strengths:**

1. The bilevel/trilevel optimization framework is mathematically coherent and provides a principled way to handle both data-based and data-free extraction.
2. The authors provide non-trivial theoretical analyses in Theorem 1 and Theorem 2 that connect defense performance with model capacity and distributional divergence.
3. The experiments are extensive and well-controlled, with multiple datasets, attacker/defender architectures, and clear ablation studies. The proposed method consistently outperforms SOTA baselines by large margins in both DFME and DBME settings.
4. Implementation details, hyperparameter ranges, and open-source code are well documented in the appendix, supporting transparency.

**Weaknesses:**

1. While the integration of data augmentation, ensemble distillation, and bilevel optimization is elegant, each element individually is well-established. The conceptual leap feels incremental rather than groundbreaking.
2. The authors are encouraged to further clarify how their proposed formulation aligns with or extends beyond the traditional OOD-based defense assumption.
3. All experiments use image classification benchmarks; no exploration is done on NLP or tabular domains, which limits generalizability.
4. Figures and algorithm boxes are dense, with limited explanation of intuition behind each step.

**Questions:**

Overall, while the paper presents a comprehensive framework with comparably solid theoretical and empirical support, the level of innovation may be more suitable for a journal extension rather than a conference contribution, given its emphasis on system integration rather than a novel conceptual breakthrough.

---

### Official Review · Reviewer_Lahy · 2025-10-30

**Soundness:** 2
**Presentation:** 2
**Contribution:** 3
**Rating:** 4
**Confidence:** 3

**Summary:**

This paper propose MISLEADER (enseMbles of dIStiLled modEls Against moDel ExtRaction), a unified and theoretically grounded defense framework for model extraction. MISLEADER uses a bilevel optimization problem to train a defense model that preserves accuracy for legitimate users while hindering an attacker’s ability to clone it. The framework employs data augmentation to simulate attacks and an ensemble of distilled models to increase robustness. Experiments show MISLEADER significantly reduces clone model accuracy across various datasets and attack types, outperforming state-of-the-art defenses while maintaining high utility.

**Strengths:**

1.Focusing on a defense that is agnostic to the query distribution is a significant and timely contribution that aligns better with real-world MLaaS deployment scenarios.

2.This paper evaluates against diverse baselines (RandP, P-poison, GRAD, MeCo, ACT, DNF) under both data-based (DBME) and data-free (DFME) attack settings, using both soft and hard labels.

3.The paper is well written and clearly structured.

**Weaknesses:**

1.This paper mentions the range and tuning of hyperparameters, but does not analyze the impact of hyperparameters on attack performance in detail.

2.Training an ensemble of models via a bilevel optimization process is significantly more expensive than training a single model or applying a lightweight output perturbation. While the paper mentions parallel deployment, a more detailed discussion on the inference-time cost would be beneficial.

3.While data augmentation is a key part of simulating attacker queries, this paper does not present a detailed ablation study on the specific choice of augmentations.

**Questions:**

Please refer to the weaknesses.

---

### Official Review · Reviewer_ddVX · 2025-11-01

**Soundness:** 3
**Presentation:** 3
**Contribution:** 2
**Rating:** 2
**Confidence:** 5

**Summary:**

This paper proposes MISLEADER, a defense mechanism against model extraction attacks. The authors target a key limitation of prior work: the reliance on an assumption that attacker queries originate from out-of-distribution (OOD) data. To address this, MISLEADER formulates the defense as a bilevel optimization problem, aiming to preserve utility on benign inputs while degrading extractability. The core of the proposed method involves using data augmentation to simulate attacker queries and deploying an ensemble of heterogeneous distilled models (e.g., ResNet, MobileNet) to increase the difficulty of cloning. The authors provide theoretical analysis for generalization and demonstrate strong empirical results, showing their method reduces clone model accuracy while maintaining or even improving utility on benchmarks like CIFAR-10/100.

**Strengths:**

Addresses a Valid Limitation: The paper clearly identifies a practical weakness in many existing model extraction defenses—their dependency on OOD detection. The goal of creating a defense that is robust to in-distribution queries is well-motivated from a security perspective.



Strong Empirical Validation: The experimental evaluation is thorough within its defined scope. The authors compare MISLEADER against a wide array of SOTA baselines across multiple datasets (MNIST, CIFAR-10, CIFAR-100) and attack scenarios (DFME, DBME). The results consistently show that MISLEADER provides a stronger defense (lower clone accuracy) while preserving high utility.





Clarity: The paper is well-written and clearly structured. The proposed method, including the optimization algorithm and ensemble strategy, is explained well .

**Weaknesses:**

Significant Practicality Concerns (Overhead): The primary weakness of this paper is the practicality of the proposed solution. MISLEADER relies on an ensemble of heterogeneous models (e.g., ResNet18_8x, MobileNetV2, DenseNet121). While this architectural diversity is key to the defense, it introduces a massive computational overhead. Both training and—more critically—inference require running multiple distinct models for every single query . For a real-world MLaaS provider, where inference latency and cost are paramount, this solution is likely non-viable. The paper fails to adequately discuss or quantify this significant trade-off.



Debatable Significance and Relevance: The paper's motivation rests on the assumption that model extraction of classic vision classifiers (like ResNets on CIFAR) is a high-priority threat for the ICLR community. This premise is questionable. The field's focus has largely shifted to large-scale generative models (LLMs, diffusion models), where the "IP" is more complex than just a set of weights for a classifier. The paper does not provide any discussion or evidence that its approach (bilevel optimization, heterogeneous ensembles) would scale to or is even relevant for these modern, high-value models. As such, the problem itself, while a classic security topic, may feel "old-fashioned" and of limited significance to the ICLR audience.


Limited Novelty: The core components of the method—model ensembling for robustness, knowledge distillation , and data augmentation —are all very well-established techniques. While their combination to solve this specific OOD-agnostic defense problem is new, the work is more of an incremental, albeit effective, engineering combination rather than a fundamental conceptual breakthrough.

**Questions:**

Overhead Analysis: Could the authors provide a concrete analysis of the inference-time overhead? Specifically, what is the increase in latency and computational cost (e.g., FLOPs) for a single prediction using MISLEADER compared to the "undefended" model and the other single-model defense baselines (like DNF or MeCo)? How can this significant cost be justified in a practical MLaaS deployment?

Scalability to Modern Architectures: The experiments are confined to small-scale vision classifiers. How is this approach expected to scale to models with billions of parameters, such as the LLMs that dominate today's MLaaS offerings? Is it feasible to train and run an ensemble of heterogeneous large-scale models, and is model extraction (as defined here) even the correct threat model for those generative systems?

---

### Note · Authors · 2025-11-23

I have read and agree with the venue's withdrawal policy on behalf of myself and my co-authors.